# Stressors, self-reported overall health, potential protective factors and the workplace well-being of nurses during the COVID-19 pandemic in Switzerland: a longitudinal mixed-methods study protocol

Claudia Ortoleva Bucher ,[1] Philippe Delmas ,[1] Annie Oulevey Bachmann ,[1] Ingrid Gilles [2]

¹La Source School of Nursing, HES-SO, University of Applied Sciences and Arts Western Switzerland, Lausanne, Switzerland
²Epidemiology and Health Systems, Center for Primary Care and Public Health, Lausanne, Switzerland

**Correspondence to**
Professor Claudia Ortoleva Bucher;
c.ortolevabucher@ecolelasource.ch

## ABSTRACT

**Introduction** The COVID-19 pandemic was making a huge impact on Europe's healthcare systems in the spring of 2020, and most predictive models concurred that pandemic waves were in the offing. Most studies adopted a pathogenic approach to the subject; few used a salutogenic approach. These showed, however, that nurses can retain their health despite a pandemic by mobilising generalised resistance resources. Our study aims to understand how nurses working in Switzerland's hospitals protected their health and workplace well-being during the COVID-19 pandemic by investigating the moderating effects of the health resources they mobilised against the stressors inherent to the situation. The study aims to explore and describe the stressors and the resources nurses used to remain healthy during the COVID-19 pandemic.
**Method and analysis** We will use a concurrent mixed-methods panel design with qualitative analyses ancillary to quantitative analyses. Quantitative data will be collected using electronic questionnaires at four time points over 2 years. Qualitative data will be collected using focus groups. Nurses from Switzerland's two main linguistic regions who had direct, indirect or no contact with patients with COVID-19 will be invited to participate. The a priori sample size will be at least 3631 participants at T0 and 1852 at T4. Longitudinal structural equation modelling and knowledge mapping will be used to analyse quantitative and qualitative data, respectively. The results derived from the two data types will then be compared and discussed using a side-by-side approach to determine whether they agree or disagree and how they complement each other to achieve our aims.
**Ethics and dissemination** Nurses will receive an electronic informed consent form. The data collected will be stored on a secure server at the authors' institution. This research project was approved by the Human Research Ethics Committee of the Canton of Vaud (2020-02845).

<div class="box">

### Strengths and limitations of this study

► The use of a salutogenic approach focusing on nurses' resources or protective health factors for dealing with stressful situations.
► The use of a mixed-methods design to enrich the examination of possible links between nurses' resources or protective health factors and workplace well-being (the study's key contribution).
► The use of a longitudinal design to examine changes in our two outcomes over time (quality of life and workplace well-being), in exposure to COVID-related stressors, in factors protective of health and in associations between them.
► The potential design adaptations needed due to uncertainties about the ongoing pandemic situation.
► The risks of attrition due to lengthy data collection.

</div>

healthcare system was very badly affected—in some countries, they were completely overwhelmed.[1 2]

Pandemic situations expose nurses to different types of stressors, such as performing unusual tasks in unusual settings (eg, units specially dedicated to the care of infected patients) and adapting to unusual work shifts. Such exposure has been associated with high levels of psychological distress[3–6] and headaches.[7] During the SARS epidemic, studies reported that between 17.3% and 75.3% of healthcare workers (HCWs) presented with mental health problems.[8–12] Two recent systematic reviews regarding SARS, together with the current literature on COVID-19, have shown that HCWs who worked in COVID-19 units were at a high risk of developing the same psychological symptoms.[3 6] Compared with other HCWs, nurses reported

## INTRODUCTION

The COVID-19 pandemic began in China in December 2019. Despite the imposition of drastic confinement measures, Switzerland's

higher stress levels, more psychopathological symptoms and more post-traumatic stress symptoms.[13 14] These problems can persist for up to 2 years after the end of an epidemic.[15] Such stressful situations generally result in greater cynicism vis-à-vis patients, more mistakes and safety issues, poorer communication among nurses and increased costs for healthcare services in terms of both human and financial resources.[16 17] Moreover, nurses are at a higher risk of burnout.[17–19] Most studies in this area have taken a pathogenic approach, focusing on identifying diseases, symptoms and risk factors.[3 4] Few studies have adopted a salutogenic approach focusing on well-being and identifying the health resources mobilised by HCWs to help them cope with their contextual stressors. Salutogenesis is 'a scholarly orientation focusing attention on the study of the origins of health and assets for health, contra the origins of disease and risk factors'.[20] It considers health as a continuum from optimal health to disease. When developing preventive actions, and instead of simply considering risk factors, salutogenesis also aims to highlight the so-called 'factors protective of health' or the 'generalised health resources' that actively support health.[20] Studies adopting this approach have shown that nurses can retain their health in such situations by mobilising 'generalised resistance resources'.[20] For example, two preliminary studies conducted in Belgium[21] and Switzerland's French-speaking region[22] have shown that the pandemic's impact has not been entirely negative and that it had a positive impact on nurses at both the personal and professional levels. In Belgium, Lecocq et al[21] conducted a qualitative study with 100 nurses working in different sectors of activity (units, medical, psychiatric units) of University Hospital of Brussels between March and June 2020. They found core themes that structured the experience of professionals during this unprecedented period. Actually, if professionals expressed 'fear and lack of safety' concerning the uncertainty of this period, they also highlighted positive aspects such as the fact to being able to engage in 'authentic relationship with patients', functioning in real support teams with less supervision from the hierarchy and 'drawing on one's resources to stay healthy'. It turns out that this period of pandemic has also afforded nurses the opportunity to highlight the added value that they bring to care and raise importance of their role in the eyes of the population.

Thus, nurses managed to find the resources to meet the challenges they faced. Gaining clearer insight into how nurses maintained their health in the face of a pandemic is essential. The salutogenic perspective has rarely been explored in a pandemic context despite its real-world applications and may provide blueprints for developing preventive interventions aimed at maintaining HCWs' health and workplace well-being during times of pandemic. Therefore, this research project will adopt a salutogenic perspective.

Most studies conducted on this topic to date have been atheoretical.[3 4] Accordingly, the relationships between concepts have varied widely. Depression, for example,

has been considered a predictor and an outcome.[3 4] To describe, explain and understand the nature and meaning of phenomena, and to develop interventions in the health field, research should be based on established models.[23] Our theoretical framework will be the Neuman systems model (NSM), considered a salutogenic model according to the criteria of both Lindström and Eriksson[24] and Mittelmark and Bauer.[20] The NSM considers health to be in a dynamic equilibrium in which human beings can retain, attain and retain stability, and in which perceived wellness is viewed as a manifestation of health.[25] Stressors are considered neutral a priori; their impact depends on circumstances, the moment and the individual's perception of them and ability to cope with them ('eustress' and 'stress').[25] An individual's ability to retain health and well-being is subject to two mechanisms: reducing exposure to stressors and activating factors protective of health. These protective factors may be physiological, psychological, sociocultural, developmental or spiritual, and they may act as moderators between stressors and health.[25 26] Finally, an individual's ability to retain their health is accordion-like: when strengthened by primary preventive interventions, it can develop and become more effective. The NSM's theoretical framework seems well suited for our study because it enables an exploration of: (1) the stressors to which HCWs' health is exposed; (2) their relationships with HCWs' overall health and well-being over time; and (3) the mechanisms used to stay healthy despite this exposure (figure 1).

Another limitation identified in studies exploring the impact of the COVID-19 pandemic or other epidemics (Severe Acute Respiratory Syndrome (SARS) or Middle-East Respiratory Syndrome (MERS)) on HCWs' health is their cross-sectional design and/or postpandemic research design.[3 4] These allowed for no examination of changes in nurses' health over time during the pandemic episode, nor for any determination of causality between exposure and outcomes. Longitudinal studies are needed to establish causal relationships between stressors, moderators and outcomes in the context of COVID-19.[27] To the best of our knowledge, there are only three longitudinal studies analysing nurses' health during COVID-19.[28–30] In Switzerland, Fuchs et al[29] conducted a longitudinal mixed-methods study, which started in April and was planned to end in December 2020, to analyse HCWs' mental health using a pathogenic perspective (outcomes include symptoms such as anxiety, depression and post-traumatic stress disorder (PTSD)). In Canada, Richardson has been monitoring HCWs for 18 months to measure their well-being and levels of distress.[28] In Portugal, Pinho et al[30] have been monitoring use of mental health promotion strategies by nurses is important to reduce stress, anxiety and depression symptoms. Our proposed longitudinal mixed-methods study using a salutogenic approach would complement these three timely studies.

Because this unprecedented pandemic situation is ongoing, with no end in sight,[27] our research project based on the NSM has the following objectives:

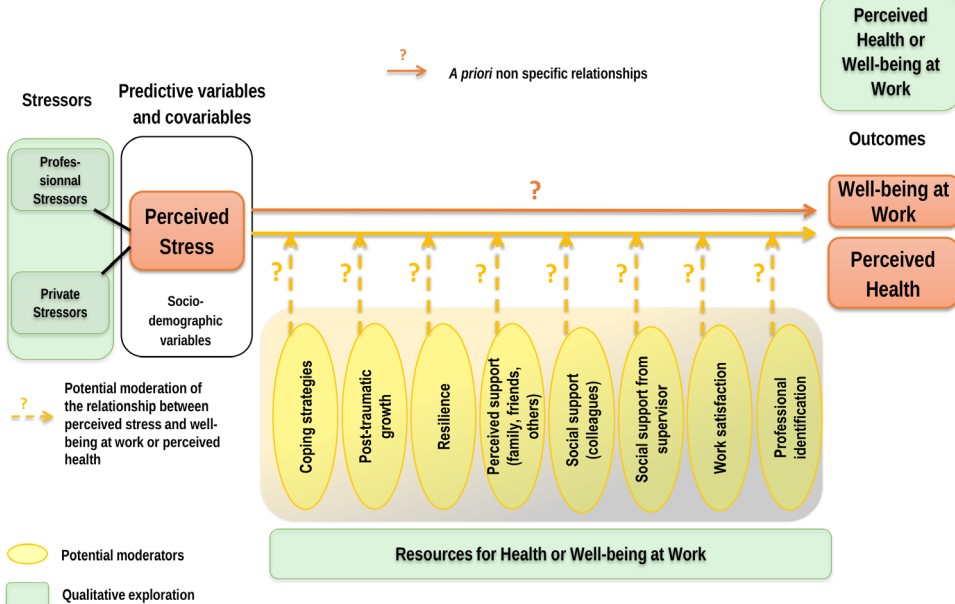

**Figure 1** Study's theoretical framework.

► To understand how nurses working in Switzerland's hospitals retained their health and workplace well-being throughout the COVID-19 pandemic (mixed methods).
► To explore the stressors experienced by nurses throughout the months of the pandemic (qualitative methods).
► To measure changes in their experience of these stressors and in any possible repercussions on their overall health and workplace well-being during the pandemic (quantitative methods).
► To explore and describe the resources used to remain healthy during the pandemic (quantitative and qualitative methods).

## METHODS AND ANALYSIS
### Study design overview
We will use a concurrent mixed-methods panel design. Quantitative analyses[31] will use data from four successive waves of self-reporting questionnaires at 6-month intervals (T0; T1 at 6 months; T2 at 12 months; and T3 at 18 months). In our study, T0 refers to the study baseline and not the time before the beginning of the pandemic. We are conscious that the situation was critical for HCWs already a year before the beginning of our study and that the study begins at a time when the resistance of the professionals has already been put to the test with a level of stress that is probably higher than usual. Actually, the preparation of the study, the construction of the study, began at the end of the first wave of COVID-19, that is, in the summer of 2020, but the important waves of the autumn pushed us to plan a beginning of the study in the beginning of 2021 in order not to solicit the professionals more at a critical period. What will be important to observe in our study will be the evolution over time of this extraordinarily extensive situation, keeping in mind

that we will not have information on the initial level of stress among nurses. This survey started in March 2021 in French-speaking Switzerland and will start in September 2021 in German-speaking Switzerland (figure 2). Qualitative interviews will take place in parallel using two focus group (FG) phases (in September 2021 and 2022) in the respective linguistic regions.

Both methodological parts are detailed below, and several reasons led to their selection (following the Strengthening the Reporting of Observational Studies in Epidemiology guidelines). Concerning the mixed-methods approach, it is recommended that multifaceted research questions use more than one method of investigation,[32] and, in our case, understanding stressors requires just such in-depth exploration. A longitudinal design will be necessary to evaluate changes over time.[33] According to Wang *et al*,[33] at least three time points are necessary to provide a robust picture of changes in sociopsychological variables and how they relate to one another. Based on the most common longitudinal wave time point intervals reported in the literature (6–12 months) (2) and our own research experience (4, 5), we chose a 6-month interval.

### Quantitative part
A self-administered electronic survey covering sociodemographic and psychological questionnaire items will be sent to nurses at 15 hospitals and clinics in the French and German-speaking regions of Switzerland by email. No sampling strategy will be applied; every eligible nurse will be contacted. A communication strategy to encourage participation has been developed (emails announcing the study, reminders). The data collection phase at each time point (T0 to T3) will last 1 month.

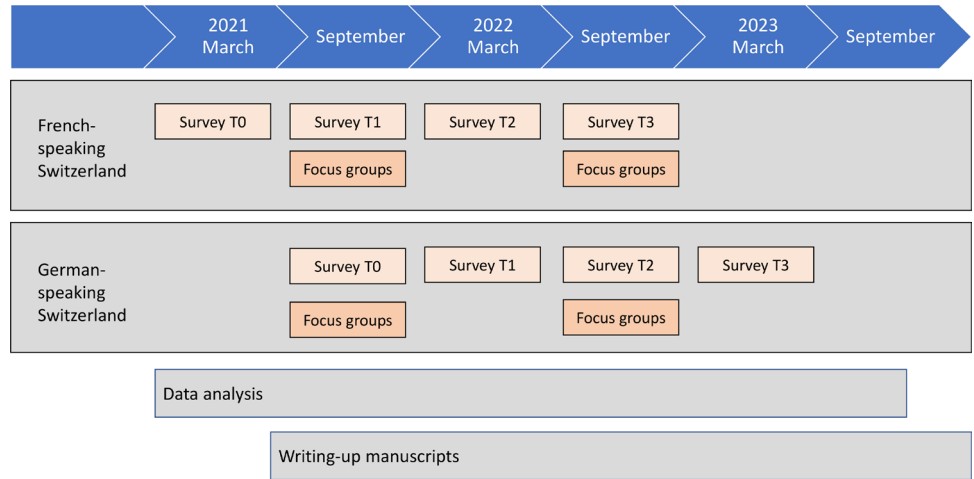

**Figure 2** Summary study timeline (March 2021 to September 2023).

## Population and sampling

The population includes all the nurses working in our partnering healthcare institutions at baseline, regardless of whether they have had direct, indirect or no contact with patients infected with COVID-19. Fifteen institutions have already agreed to facilitate the study, making our potential population 5187 nurses working in different hospitals in Switzerland's German and French-speaking regions.

Inclusion criteria at baseline are: (1) having a long-term (not temporary) contract at the institution selected; (2) working at least 50% of a full-time equivalent position; (3) being able to read and understand French or German; and (4) having signed the written informed consent form. Exclusion criteria at baseline are: (1) being a nurse manager; (2) not working at the institution during the COVID-19 crisis; and (3) being a student nurse during the COVID-19 crisis.

## Recruitment targets

We used previous relevant studies to estimate participation and attrition rates. A recent Swiss survey of HCWs' overall health had a response rate of 37%.[22] Concerning attrition rates, Niu et al[34] expect 20% in their study of nurse sleep quality and Kovner et al[35] reported a 30% attrition rate in their panel study of nurse turnover. The Swiss Household Panel survey recorded attrition rates from 13% to 18%.[36] We therefore estimate a 20% attrition rate at each data collection time point. Thus, we expect our accessible population of 5187 nurses at T0 to fall to 3631 at T1 (80% of T0 staff), 2905 at T2 (80% of T1 staff), 2324 at T3 (64% of T1 staff), with a final sample of 1852 at T4 (51% of T1 staff).

## Measurement

The questionnaire is composed of the following scales.

### Predicting variable

The Perceived Stress Scale,[37] in validated French (Cronbach's α=0.78–0.87) and German versions (Cronbach's α=0.84),[38 39] is a 10-item scale to evaluate the degree to which life situations are generally perceived as threatening, that is, as unpredictable, uncontrollable and painful.[40]

### Outcome variables

The Psychological Well-Being Scale, developed by Diener et al[41] and adapted for the workplace by Fisher,[42] in validated French and German versions (Cronbach's α=0.88 and confirmed the scale's one-dimensional structure),[43–45] is an 8-item scale to evaluate workplace well-being and measures self-perceived functioning in areas such as self-esteem, purpose and relationships. Convergent validity analyses have shown good correlations (0.43>r<0.73) with the Basic Needs Satisfaction Scale,[46] Ryff's Psychological Well-being Scale,[47] the Satisfaction with Life Scale[48] and the Life Orientation Test-Revise.[43]

The WHO Quality of Life-BREF (WHOQOL-BREF),[49] in validated French and German versions,[50 51] is a 26-item short version of the 100-item WHOQOL scale covering four domains—physical health, mental well-being, social relations and work environment—that measure self-perceived quality of life. Psychometric properties have proved good to excellent for the original version,[49] and the French and German versions showed Cronbach's α>0.65 for all dimensions.[50 51]

### Mediating variables

The Brief Coping Orientation to Problems Experienced Inventory (Brief-COPE),[52] in validated French and German versions,[53 54] is a 28-item scale evaluating several coping strategies. The French and German subscales have obtained Cronbach's α=0.50–0.90 (total score is not representative).

### Moderating variables

The Post-Traumatic Growth Inventory (PTGI)-Short Form,[55] in validated French and German versions,[56–58] is a 10-item inventory evaluating post-traumatic growth, that is, positive psychological change experienced following a traumatic event. The PTGI has demonstrated good internal coherence (Cronbach's α: French version: 0.90 for total score) and acceptable test–retest (r=0.71).

The 10-item Connor-Davidson Resilience Scale,[59] in validated French and German versions,[59 60] measures a person's ability to bounce back when confronted with the difficulties that may arise in life. This 10-item unidimensional version possesses excellent psychometric properties, better than those of the original 25-item version, and its use is suited to large-scale epidemiological studies.[59 61 62] Its 3-month test–retest is good.[63]

The Multidimensional Scale of Perceived Social Support,[64] in validated French and German versions,[65–67] is a 12-item scale measuring perceived social support from family, friends and significant others. Cronbach's α range from 0.91 to 0.94 for the French and German versions.[65–67] Its 4-month test–retest has ranged from 0.72 to 0.85.[64]

The Copenhagen Psychosocial Questionnaire (COPSOQ),[68] in validated French and German versions,[69 70] is a comprehensive tool for assessing psychosocial risk in the workplace through 24 core dimensions covering four aspects of work: work environment, health, well-being and personality. We will use three of these dimensions: dimensions of social support from colleagues, dimensions of social support from supervisors and dimensions of satisfaction with job quality. The COPSOQ has been used in various work contexts in different languages, including French[69] and German,[70] and has been constantly improved, the latest version having been proposed by its authors in 2018.[71] These dimensions have good internal consistency (all α>0.80).

Professional identification will be measured using the single-item measure of social identification proposed by Postmes et al,[72] which can be adapted to any social group, including professional ones. Its simple wording ('I identify with [target group]') reduces interpretation bias and translation issues; its 7-point rating scale ranges from 1=fully disagree to 7=fully agree. The tool has been tested against more classic scales such as the 5-item Social Identity Scale[73] and the 14-item Leach Scale[74] and showed good convergent, divergent and test–retest validity. Moreover, a meta-analysis found this single-item measure to possess stronger reliability compared with other 'tried-and-trusted' single-item measures.[72]

Sociodemographic variables. Information regarding gender, age, family situation, and number of children, employment rate, years since last diploma, years of experience in current department, level of exposure to COVID-19 (nurses working in COVID-19 care units; nurses working in institutions that admitted infected patients, but not in COVID-19 units; nurses working in institutions that did not admit infected patients) and continuing professional development programme completed will be collected. As suggested by Aiken et al,[75] hospitals' organisational indicators (eg, nurse-patient ratios, work schedules, average time spent with patients per shift) will be collected at each measurement point.

## Quantitative data analyses

Standard data quality checks will be performed to remove unreliable values from the database, and descriptive statistics will be computed to qualify those data.

As specified in the Introduction section, the NSM is used as conceptual framework in this study. According to this model, the ability of individuals to maintain their health and well-being against a priori neutral stressors is conditioned on the mobilisation of protective factors. It has thus been proposed that some factors could moderate the negative association between stressors and the normal line of defence what would result in an optimal health. However, the mechanisms underlying the action of protective factors as well as the identification of these factors are still discussed.[26] Concerning the identification of factors, the underlying postulate of the NSM is that they vary according to the situation. That is why we selected relevant variables on the basis of the existing literature on previous outbreaks, but also on the basis of studies using a salutogenic approach specifically in the COVID-19 context. For example, resilience, social support (including this provided by colleagues and supervisors), post-traumatic growth and identification to professional group appeared to be key resources for nurses during critical situations.[12 15 76–83] The aim of quantitative analyses will be thus to identify, in a longitudinal perspective, which factors were of most importance during the pandemic to buffer the effect of stressors faced by nurses on their health and professional well-being (our outcomes). To reach these objectives, association between perceived stress and well-being will subsequently be modelled using mixed-effects regressions. Random effects are necessary to model the data's longitudinal structure. Linear models will be the default choice, but we will not rule out resorting to data manipulation (eg, dichotomisation) or alternative models (eg, logistic models) if the data's structure requires it. The moderating effects of post-traumatic growth, resilience, perceived support, coping strategies, job satisfaction and professional identity will be evaluated by entering the estimated interactions between these variables and perceived stress into the regression models. A frequentist probability approach will be used for all our analyses. All calculations will be run on the latest available stable version of R software. Statistical significance will be set at p value <0.05.

## Qualitative part

We will use a knowledge mapping approach.[84] Mind mapping is a useful graphical format for representing key themes raised during FGs. It can help stimulate and galvanise discussions and keep them on track, enhance transparency and group ownership of the data analysis process and enable a rapid dynamic to develop between data collection and feedback.[85] This pragmatic approach is considerably faster than traditional methods used for analysing FGs, but it produces broadly similar results.[85] Participants are actively involved in interpreting the resulting mind maps.[86] The advantages of mapping

include its 'free form' and unconstrained structure, meaning there are no limits to the ideas and links that can be made and no need to retain an ideal structure or format. Mapping thus promotes creative thinking and encourages brainstorming. The three steps to structuring a knowledge mapping approach are: (1) developing interview guidelines, (2) conducting FG interviews, and (3) analysis and conclusion.[84]

## Population

The target population will consist exclusively of nurses who respond to the questionnaire at T0 and indicate their interest in participating in the study's future stages by giving their written informed consent at that time. The interested nurses in each linguistic region will be divided into three groups according to their degree of contact with patients with COVID-19: (1) nurses in direct contact with infected patients (worked in COVID-19 care units); (2) nurses in indirect contact because their institution admitted infected patients, but they did not work in COVID-19 care units; (3) nurses with no contact because their institutions did not admit infected patients.

## Sampling and recruitment targets

The research team will select three samples of nurses to participate in FGs in each of the two linguistic regions 12 months apart. Nurses will be allowed to participate in the FGs during their paid work hours. If there are too many volunteers, participants will be selected using a convenience sampling method based on sociodemographic characteristics such as age, their institution's geographical location, whether their COVID-19 experience had been on their usual ward and to foster heterogeneity among the FG participants. If there has been attrition in the FGs at T3, groups will be completed with new participants selected in the same way.

## Data collection

Investigators began by developing a semistructured interview guide based on the NSM and tested it on two nurses not participating in the study. Several pathways will be explored: lived experiences and stressors, health, well-being at work, as well as the evolution of these components over time and the resources mobilised to retain them.

FG discussion will last about 1.5 hours and will be held by videoconference that facilitates mutual trust and ensures confidentiality. Participants will be formally asked for their consent before conversations are recorded. A moderator (a senior researcher) will lead each FG, assisted by a scientific collaborator. Both will be trained beforehand and will receive precise guidelines to ensure standardised procedures. All the topics in the interview guide will be discussed.

A knowledge map of each topic discussed will be drawn on a flip chart, resulting in a synthesis formed by a complex network of representations in text: 'A knowledge map is the visual display of captured information and relationships [...]'.[87] Central concepts within each topic will be arranged in relation to one another, and recurring concepts will be highlighted. Before moving from one topic to the next, the moderator will ensure that every input has been documented and that participants validate the map to ensure that the researchers have properly understood each point as they have been expressed.[84] The knowledge maps will also be considered as protocols for the FG discussions.

## Qualitative data analysis

Qualitative analysis will begin during the FG sessions when participant feedback confirms that each map fully summarises each topic's discussion. Knowledge maps will then be digitised and merged by interview topic. They will then be translated into English to enable the complete integration of the different German and French language maps. The integrated maps in English will then be discussed, and the qualitative research team will cluster themes. Each cluster will be described with regard to the study's aims.

To ensure rigour in the qualitative data collection process, the investigators will maintain a logbook of methodological, theoretical and personal notes.

## Integrating quantitative and qualitative data

This research is based on a concurrent mixed design (QUANTI/Quali). At T1 and T3, qualitative and quantitative data will be collected in parallel. The qualitative data are ancillary to the quantitative data. The interview guide for collecting the former will therefore not be dependent on the results of the latter.

The qualitative data will shed light on the experience of the crisis at T1 and T3, its links with perceived health and well-being at work overtime, and the QUANTI health maintenance measures will highlight the nurses' use of protective factors/health resources during the pandemics.

The integration of the data (QUANTI/Quali) will be done through a process of comparison on two occasions (T1 and T3). For this purpose, we plan to bring together the QUANTI and Quali research teams from both language regions in workshops. Similarities and differences between QUANTI and Quali results will be investigated and discussed. The mixed product will consist of interpretations or decisions developed in the research team that take into account the explicit interdependencies between QUANTI and Quali data.[31]

## Patient and public involvement

Patients and/or the public were involved in the design, conduct, reporting or dissemination plans of this research. Refer to the Methods and analysis section for further details.

## Study status

Participant recruitment for the survey's first round (T0) began in March 2021 in French-speaking Switzerland. Subsequent rounds are planned for September 2021

(T1), March 2022 (T2) and September 2022 (T3). Participation in German-speaking Switzerland will occur with a 6-month delay: T0 in September 2021, T1 in March 2022, T2 in September 2022 and T3 in March 2023. This time lag will be used to collect qualitative data. Thus, the FGs for French and German-speaking Switzerland are scheduled for September 2021 and September 2022, respectively.

## Ethics and dissemination

This project was approved by the Human Research Ethics Committee of the Canton of Vaud (project number: 2020-02845). Each partner institution will designate a reference person. The research team will send that person an anonymous link to the online questionnaire and an information letter and a standard study invitation email to be adapted for each institution.

At T0, each reference person will send out their adapted information email to their institution's potential participants (specifying that the study is longitudinal and includes four waves). These nurses will be free to accept or decline participation. At the end of the questionnaire, participants who wish to join in the next waves and FGs will be asked to provide an email address so the research team can contact them directly.

Participants will be able to withdraw from the study at any time. Email addresses will be kept by the research team member responsible for the logistical aspects of data collection (sending invitation emails and reminders). Other research team members will not have access to this information. Nominative information (eg, emails) will be deleted at the end of the study.

Participation in both the quantitative and qualitative parts of this study will be voluntary—nurses will be under no obligation. Electronic 'clicked' consent will be collected from all participants in the quantitative part on a form stating that their data may be used and shared with other researchers in an anonymised form and that they will not be identified nominatively in any outputs. All participants will have a unique anonymous identifier, and only this identifier will appear in the data sets. Separate files with personal details will be kept securely, will not be shared with other researchers and will be deleted on completion of the project. At least 72 hours before collecting qualitative data in FGs, participants will receive an information letter and a consent form for signature. Recordings will not be shared, and transcriptions will be anonymised. Personal data enabling the identification of participants will be kept separate from survey responses and will be deleted when data collection is complete. Any data to be made public will be entirely anonymous.

Nurses will be under no obligation to answer any or all of the questions, and they will be free to interrupt questionnaires at any time. In this event, answers already provided will not be registered.

Results will be disseminated in several ways. The first persons to benefit from this research will be the participants themselves. A document containing the study's highlights will be prepared specifically for them and made available on our partner institutions' websites or on demand. These institutions will hold conferences open to the public. Scientific publications will be addressed to the French, English and German-speaking scientific communities (eg, *BMC Health and Services Research, Journal of Clinical Nursing, International Journal of Nursing Research, Recherche en Soins Infirmiers, etc*). Results will also be disseminated at international and local conferences. Finally, a research report containing the study's key findings will be produced for the libraries of the different partnering institutions and for the funding partner.

## Study significance

We believe that this study will make a notable contribution to advancing scientific knowledge on HCWs' health and workplace well-being during pandemics, particularly on the following aspects: (a) analysing changes in the effects of various potential factors protective of nursing health and workplace well-being; (b) developing knowledge using an established theoretical framework in a novel context; and (c) guiding the development of new preventive interventions to support nurses in times of pandemic. Several elements are worthy of note regarding the potential benefits of this study for our institutional field partners. Study findings could be integrated into quality improvement processes at participating institutions and could inform the development of support programmes there for preventing disease and promoting health among their HCWs.

Embedding the NSM's theoretical framework in a salutogenic approach[25] will provide us with an essential insight into what protects nurses' overall health and workplace well-being during a pandemic such as the one we are living through. It will shed light on the different stressors that they are exposed to. This is a vital step towards developing preventive interventions geared both to mobilising factors protective of health and to reducing exposure to risk factors. Such interventions would support nurses' overall health and workplace well-being.

### Broader impacts

Several elements are worthy of note regarding the potential benefits of this study for our institutional partners. The findings could be integrated into their quality improvement processes and could inform the development of support programmes for preventing disease and promoting health among their staff, particularly nurses.

The results of this research could also be integrated into undergraduate nursing curricula or even continuing professional development programmes at partnering institutions or in healthcare management education programmes in Switzerland. More broadly, they will also support the healthcare system's performance when faced with crises of a similar type.

**Contributors** All authors contributed equally to the conceptualisation of the study. All authors are involved in the data recruitment. IG hosted the survey and all authors are involved in the data analyses. All authors significantly contributed to the writing

of the manuscript. The manuscript was reviewed and edited prior to submission, and all authors agreed on the final version.

**Funding** The study is supported by the Swiss National Science Foundation (10001C_201137/1).

**Competing interests** None declared.

**Patient and public involvement** Patients and/or the public were not involved in the design, or conduct, or reporting, or dissemination plans of this research.

**Patient consent for publication** Not required.

**Ethics approval** This project was approved by the Human Research Ethics Committee of the Canton of Vaud (project no. 2020-02845)

**Provenance and peer review** Not commissioned; externally peer reviewed.

**ORCID iDs**
Claudia Ortoleva Bucher http://orcid.org/0000-0002-8411-4181
Philippe Delmas http://orcid.org/0000-0001-8169-470X
Annie Oulevey Bachmann http://orcid.org/0000-0001-9537-3587
Ingrid Gilles http://orcid.org/0000-0003-2051-4749

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
