## [Reviewer comments · BMJ Open]

ARTICLE DETAILS

TITLE (PROVISIONAL)	Stressors, self-reported overall health, potential protective factors and the workplace well-being of nurses during the COVID-19 pandemic in Switzerland: a longitudinal mixed-methods study protocol
AUTHORS	Ortoleva Bucher, Claudia; Delmas, Philippe; Oulevey Bachmann, Annie; Gilles, Ingrid

VERSION 1 – REVIEW

REVIEWER	Wilbiks, Jonathan University of New Brunswick, Psychology
REVIEW RETURNED	29-Sep-2021

GENERAL COMMENTS	This protocol provides a well thought out explanation of a study that will investigate the well-being of health care workers in Switzerland during the ongoing COVID-19 pandemic. In my assessment, this protocol is clear, well-reasoned, and should produce results that can be analysed and applied to answer the research questions. That being the case, the points provided here are intended to provide avenues for the authors to revise the proposal in order to make it more clear in certain areas, as well as to make more specific predictions and analysis plans. On p.4, the authors state "...pandemic's impact has not been entirely negative". I would urge the authors to make very clear what they mean by this statement, as I think many would take exception to it. This survey study started in March 2021 (deemed "Time 0" for the longitudinal study), but this is NOT Time 0 for COVID-19, it is approximately one year after COVID became an issue in Europe. This should be acknowledged and explained - specifically the fact that there was ALREADY likely a large amount of mental health / burnout issues in the health care worker populations as of the beginning of this study. I understand it is too late to do this retroactively, but it would be of use to include a scale measuring burnout syndrome in addition to the other scales being employed. For each scale being used, the authors should provide information about reliability and validity of the scale (preferably being validated on health care worker populations). For the quantitative data analysis, the authors should specify their expected effects for their analysis models. Right now they say they
---

	will be using linear models, and looking at a long list of moderators. It is my opinion that the moderators being used should be chosen based on theoretical perspectives and previous findings, and should be specified a priori. I do not understand the description of how the quantitative and qualitative data will be integrated. I would urge the authors to expand this section in order to clarify this. There are typographical errors sprinkled through the paper. For example, p.4 l.27 "cynism" should be cynicism P.9 L.21 - alpha = .95 should read alpha = .05 (I hope)
--	---

REVIEWER	de Pinho, Lara Manuela University of Évora São João de Deus Higher School of Nursing
REVIEW RETURNED	30-Oct-2021

GENERAL COMMENTS	Congratulations to the authors on their excellent study. The protocol is well written and reasoned. However, there are some improvements to be made. In this sentence "To the best of our knowledge, there are only two ongoing longitudinal studies analyzing nurses' health during COVID-19". This is not true. There is a longitudinal study in Portugal on mental health in nurses and the use of mental health promotion strategies. Please see this study: https://doi.org/10.1016/j.envres.2021.110828 Methods: The authors should mention which methodological model they used to write the protocol, following all the steps of this model.
--

VERSION 1 – AUTHOR RESPONSE

Reviewer: 1

Dr. Jonathan Wilbiks, University of New Brunswick Comments to the Author:

This protocol provides a well thought out explanation of a study that will investigate the well-being of health care workers in Switzerland during the ongoing COVID-19 pandemic. In my assessment, this protocol is clear, well-reasoned, and should produce results that can be analysed and applied to answer the research questions. That being the case, the points provided here are intended to provide avenues for the authors to revise the proposal in order to make it more clear in certain areas, as well as to make more specific predictions and analysis plans.

Response: Thank you for your comments.

On p.4, the authors state "...pandemic's impact has not been entirely negative". I would urge the authors to make very clear what they mean by this statement, as I think many would take exception to it.

Response: We thank you for your remark. We understand your concerns and we indeed agree that crises, such as the one we are currently experiencing, have many negative impacts on professionals and the population. Nevertheless, these crises lead people to mobilize coping resources. During these phases of adaptation, individuals can learn from what they have mobilized facing these harsh

situations to better function in future similar situations. Actually, studies from Belgium and Switzerland support the idea of a beneficial effect. We have therefore added a paragraph reporting these results to illustrate our point.

The following paragraph has been added:

“For example, two preliminary studies conducted in Belgium (21) and Switzerland’s French-speaking region (22) have shown that the pandemic’s impact has not been entirely negative and that it had a positive impact on nurses at both the personal and professional levels. For example, in Belgium, Lecocq et al. (23) conducted a qualitative study with 100 nurses working in different sectors of activity (units, medical, psychiatric units) of University Hospital of Brussels between March and June 2020. They found core themes that structured the experience of professionals during this unprecedented period. Actually, if professionals expressed “fear and lack of safety” concerning the uncertainty of this period, they also highlighted positive aspects such as the fact to being able to engage in “authentic relationship with patients”, functioning in real support teams with less supervision from the hierarchy and “drawing on one’s resources to stay healthy”. It turns out that this period of pandemic has also afforded nurses the opportunity to highlight the added value that they bring to care and raise importance of their role in the eyes of the population.”

This survey study started in March 2021 (deemed "Time 0" for the longitudinal study), but this is NOT Time 0 for COVID-19, it is approximately one year after COVID became an issue in Europe. This should be acknowledged and explained - specifically the fact that there was ALREADY likely a large amount of mental health / burnout issues in the health care worker populations as of the beginning of this study.

Response: Indeed, T0 refers to the study baseline.

We added the following sentence to specify this point:” In our study, T0 refers to the study baseline and not the time before the beginning of the pandemic. We are conscious that the situation was critical for health care workers already a year before the beginning of our study and that the study begins at a time when the resistance of the professionals has already been put to the test with a level of stress that is probably higher than usual. Actually, the preparation of the study the construction of the study began at the end of the first wave of COVID-19, i.e., in the summer of 2020, but the important waves of the autumn pushed us to plan a beginning of the study in the beginning of 2021 in order not to solicit the professionals more at a critical period. What will be important to observe in our study will be the evolution over time of this extraordinarily extensive situation, keeping in mind that we will not have information on the initial level of stress among nurses.”

I understand it is too late to do this retroactively, but it would be of use to include a scale measuring burnout syndrome in addition to the other scales being employed.

Response: We thank you for your suggestion. Actually, as we’re using a salutogenic approach, we did not include a burnout scale because we were looking for factors protective of health and their potential interaction on the relationship between exposure to stressors and overall health or workplace wellbeing. Moreover, we are currently collecting data for the 2nd wave of the quantitative phase of the study. It is thus too late to include a new scale.

For each scale being used, the authors should provide information about reliability and validity of the scale (preferably being validated on health care worker populations).

Response: Information on psychometric properties of all scales were added in the Measurement section.

For the quantitative data analysis, the authors should specify their expected effects for their analysis models. Right now they say they will be using linear models, and looking at a long list of moderators. It is my opinion that the moderators being used should be chosen based on theoretical perspectives and previous findings, and should be specified a priori.

Response: as specified in the introduction section, the Neuman Systems Model (NSM) is used as conceptual framework in this study. According to this model, the ability of individuals to maintain their health and well-being against a-priori neutral stressors is conditioned on the mobilization of protective factors. It has been proposed that some factors could moderate the negative (buffer effect) association between stressors and the normal line of defense what would result in an optimal health. However, the mechanisms underlying the action of protective factors as well as the identification of these factors are still discussed. In our study protective factors are resilience, post-traumatic growth, social support (including at the professional level from colleagues and hierarchy) and professional identification. The moderating effects of post-traumatic growth, resilience, perceived support, coping strategies, job satisfaction and professional identity will be evaluated by entering the estimated interactions between these variables and perceived stress into the regression models. The aim of quantitative analyses will be thus to identify which factors were of most importance during the pandemic to buffer the effect of stress on nurses' health and professional well-being (our outcomes).

Concerning the variables, as the NSM does not identify specific protecting factors (they can vary according the context) those included in our study were chosen according to the literature existing on previous outbreaks, but also on the bases of the studies already existing on the COVID-19 context and which used a salutogenic approach. To clarify this with changed the status of our variables is now specified in the measure section and more details about analyses have been added in analyses description : "As specified in the introduction section, the Neuman Systems Model (NSM) is used as conceptual framework in this study. According to this model, the ability of individuals to maintain their health and well-being against a-priori neutral stressors is conditioned on the mobilization of protective factors. It has thus been proposed that some factors could moderate the negative association between stressors and the normal line of defence what would result in an optimal health. However, the mechanisms underlying the action of protective factors as well as the identification of these factors are still discussed (Gigliotti, 2012). Concerning the identification of factors, the underlying postulate of the NSM is that they vary according to the situation. That is why we selected relevant variables on the basis of the existing literature on previous outbreaks, but also on the basis of studies using a salutogenic approach specifically in the COVID-19 context. For example, resilience, social support (including this provided by colleagues and supervisors), post-traumatic growth and identification to professional group appeared to be key resources for nurses during critical situations (15 77-85). The aim of quantitative analyses will be thus to identify, in a longitudinal perspective, which factors were of most importance during the pandemic to buffer the effect of stress on nurses' health and professional well-being (our outcomes) by strengthening coping strategies. To reach these objectives, association between..."

I do not understand the description of how the quantitative and qualitative data will be integrated. I would urge the authors to expand this section in order to clarify this.

Response: We brought changes to the paragraph in order to make it clearer and explain how the integration of both quantitative et qualitative data will be handled. The following paragraph had been added: "This research is based on a concurrent mixed design (QUANTI/quali). At T1 and T3 qualitative and quantitative data will be collected in parallel. The qualitative data are ancillary to the

quantitative data. The interview guide for collecting the former will therefore not be dependent on the results of the latter.”

The qualitative data will shed light on the experience of the crisis at T1 and T3, its links with perceived health and well-being at work overtime and a possible dynamic in the nurses' use of protective factors/health resources with the QUANTI health maintenance measures.

The integration of the data (QUANTI/quali) will be done through a process of comparison on two occasions (T1 and T3). For this purpose, we plan to bring together the QUANTI and Quali research teams from both language regions in workshops. Similarities and differences between QUANTI and Quali results will be investigated and discussed. The mixed product will consist of interpretations or decisions developed in the research team that take into account the explicit interdependencies between QUANTI and Quali data (Creswell and Plano Clark 2011).

There are typographical errors sprinkled through the paper. For example, p.4 l.27 "cynism" should be cynicism

Response: Changes have been made

P.9 L.21 - alpha = .95 should read alpha = .05 (I hope)

Response: p-value < .05 was added.

Reviewer: 2

Dr. Lara Manuela de Pinho, University of Évora São João de Deus Higher School of Nursing, Comprehensive Health Research Centre (CHRC)

Comments to the Author:

Congratulations to the authors on their excellent study. The protocol is well written and reasoned. However, there are some improvements to be made. In this sentence "To the best of our knowledge, there are only two ongoing longitudinal studies analyzing nurses' health during COVID-19". This is not true. There is a longitudinal study in Portugal on mental health in nurses and the use of mental health promotion strategies. Please see this study: <https://doi.org/10.1016/j.envres.2021.110828>

Response: Thank you for your comments. We have now added the reference of the longitudinal study in Portugal.

Methods:

The authors should mention which methodological model they used to write the protocol, following all the steps of this model.

Response: The statement “following the STROBE guidelines” was added.

VERSION 2 – REVIEW

REVIEWER	de Pinho, Lara Manuela University of Évora São João de Deus Higher School of Nursing
REVIEW RETURNED	16-Nov-2021
GENERAL COMMENTS	Congratulations for you work!